# Impacts of National Drug Price Negotiation on Expenditure, Volume, and Availability of Targeted Anti-Cancer Drugs in China: An Interrupted Time Series Analysis

**DOI:** 10.3390/ijerph19084578

**Published:** 2022-04-11

**Authors:** Yan Sun, Zheng Zhu, Jiawei Zhang, Peien Han, Yu Qi, Xiaoyang Wang, Li Yang

**Affiliations:** Department of Health Policy and Management, School of Public Health, Peking University, Beijing 100191, China; sunyanedu@126.com (Y.S.); 1710306118@pku.edu.cn (Z.Z.); jiaweizhang1997@126.com (J.Z.); 1610306118@pku.edu.cn (P.H.); taianqy@sina.com (Y.Q.); wangxiaoyang2young@163.com (X.W.)

**Keywords:** national price negotiation, interrupted time series, targeted anti-cancer drugs, China

## Abstract

The Chinese government has launched six rounds of national drug price negotiation since 2016 to lower the price and expand access to innovative drugs, many of which are anticancer drugs. This study aims to examine the effect of the second round of negotiation at the provincial level on the expenditure, volume, and availability of anti-cancer drugs. Procurement data at the provincial level from January 2017 to September 2018 were extracted from the China Drug Supply Information Platform (CDSIP). The volume, expenditure, and availability of three targeted anti-cancer drugs, rituximab, trastuzumab, and recombinant human endostatin (RHE), in 11 provinces that implemented the policy in September 2017 were analyzed through a controlled interrupted time series (ITS) analysis. A significant 6.0% increase (*p* < 0.1) in monthly average expenditure, an increase in the volume of 99.51 DDDs (defined daily doses) (*p* < 0.1), and a 0.24% (*p* < 0.1) increase in availability were observed for rituximab following the implementation of the policy. The volume and availability of rituximab increased by 949.6 DDDs (*p* < 0.05) and 1.56%, respectively, immediately after implementation. The availability of trastuzumab increased by 5.14% (*p* < 0.01) immediately after the implementation while no instant changes in expenditure and volume were observed. A 15% (*p* < 0.01) increase in monthly expenditure, 3673.17 DDDs increase in volume, and 0.66% increase in availability were observed after the inclusion of Trastuzumab. However, for RHE, only a 0.32% (*p* < 0.01) increase was observed after its inclusion. Eastern and middle provinces benefited more than western provinces. National negotiation related to the drug price significantly increased the volume and expenditure of anti-cancer drugs and improved their availability. The effect of the policy might be different across different regions and across different anticancer drugs.

## 1. Introduction

Cancer is one of the major public health challenges in the world [1]. In China, cancer was the leading cause of death in urban areas (25.4%) and the third leading cause of death in rural (23.0%) areas in 2020 [2]. The launch of targeted anticancer drugs with potentially lower normal tissue toxicity and higher efficacy brings hope for cancer patients but, at the same time, their high prices and expanding expenditure also poses challenges for payers worldwide [3,4]. It is estimated that the expenditure of anticancer drugs increased from 96 billion dollars in 2016 to 164 billons dollars in 2020 and is projected to grow to 269 billion dollars by 2025 [5].

In China, affordability and availability of anticancer drugs are also major challenges. A multi-center cross-sectional survey in 2012–2014 suggested that average expenditure for treating cancer in China was $9739 per patient while the average annual household income was only $8607 [6]. The price of anti-cancer drugs for Chinese patients is high relative to their income. A comparison of prices for eight anticancer drugs in seven countries revealed China has the second-worst affordability [7]. In addition, fewer innovative drugs were approved and longer waiting times were needed for innovative to enter medical insurance catalogues compared with other countries and the Asia-Pacific region. For example, the average waiting time for entering reimbursement lists in China of the 36 innovative medicines in the second round of negotiation was 54 months, which is far longer than that in Japan (two months), Australia (16 months), and Taiwan of China (14 months) [8]. 

Tendering and negotiation are widely applied approaches for both developing and developed countries to obtain price discounts for drugs, especially in the reimbursement decision-making process [9,10,11]. In the Asia-Pacific region, negotiation usually happens simultaneously with tendering to determine details such as the volume and supplier of drugs with existing competition [9]. However, some countries use negotiation as a strategy to include innovative drugs for reimbursement with a price discount, such as China [12] and South Korea [13,14]. Some evidence revealed that negotiation might not be an effective method regarding price with few competitors [10], but in China, the average price discount for newly added drugs exceeded 50% from 2018 to 2021. China has implemented six rounds of national drug price negotiation since 2016 to introduce innovative and unaffordable medicine into the national reimbursement list and included 3, 36,17, 70, 96, and 67 drugs in the national reimbursement drug list [12,15,16] (see Appendix A). The 2016 negotiation could be seen as a pilot and only included three drugs while the 2017 negotiation first established a framework for following negotiations, which consist of preparation (issuing relative rules and regulations and establishing expert databases for evaluation), evaluation (soliciting advice and opinion and confirming negotiated drugs based on experts’ voting), negotiation, and implementation.

Some studies have concluded the background and procedure of negotiations and projected the impact on reducing the financial burden and encouraging domestic innovation [12,17,18]. Recent studies investigating the impact of the first or second round of negotiation in China have shown that negotiation effectively reduced the cost to patients, increased the volume, and had no significant change or decreased hospital spending [19,20]. Analysis in Nanjing, a city in Jiangsu Province, revealed that policy increased the utilization of anticancer drugs but the availability remained low [21]. At the same time, the development of domestic me-too drugs can further aid price reduction by increasing competition [22]. 

However, some reports questioned the effect of the policy that under the limit on the total budget and drug expenditure proportion, hospitals may not provide these innovative drugs for patients, especially anti-cancer drugs because of their relatively high cost. As a consequence, patients have to purchase these drugs through other channels and without the reimbursement of health insurance [21,23]. Current literature focuses on the impact of policy on sample hospitals [19,20] or hospitals in a province [21], and no research on nation-wide availability has been conducted, with limited research on overall expenditure at the provincial level. Given this circumstance, this article aims to fill the gap and assess the impact of the second round of national negotiation on the expenditure, volume, and availability of anticancer drugs at a provincial level in China. 

## 2. Materials and Methods

### 2.1. Data Sources

The procurement data of 36 kinds of negotiated drugs from January 2017 to September 2018 were extracted from the China Drug Supply Information Platform (CDSIP). CDSIP is a national platform collecting and gathering drug procurement data from all centralized drug procurement platforms at a provincial level through Yao Pin Identification (YPID) coding, an identical code for the strength of the medicine [24]. The Chinese government began establishing CDSIP in 2015 and completed it in December 2017. The generic name, strength, dosage form, average procurement price, procurement amount, and the number of hospitals that procured certain types of drugs were extracted.

### 2.2. Data Selection

We chose provinces that had fewer than two months of missing data from January 2017 to September 2018. In our primary analysis, most provinces (23 provinces) implemented the negotiation price in September 2017. Eleven provinces (Anhui, Beijing, Guangxi, Henan, Jilin, Hebei, Shaanxi, Chongqing, Zhejiang, Fujian, Hubei) that implemented the policy in September 2017 and with fewer than two months of missing data were selected. 

A total of 15 targeted anti-cancer drugs were identified among 36 negotiated drugs [19]. Considering the data availability, we chose three anti-cancer drugs with relatively high average procurement expenditure, rituximab, trastuzumab, and recombinant human endostatin, as the intervention group. Following the guidance proposed by Bernal et al. [25] and considering the data availability, we chose Pegaspargase as a comparison group because it has no common indications with the three targeted anticancer drugs in the intervention group and thus would not have a substitution effect. Table 1 illustrates the detailed information of the four anti-cancer drugs. 

### 2.3. Statistical Analysis

A controlled interrupted time series analysis was conducted to explore the effect of intervention on the procurement expenditure, procurement, and availability of the anti-cancer drugs. According to World Health Organization (WHO), availability is defined as “the percentage of medicine outlets in which the medicine was found on the day of data collection” [26].

Interrupted time series analysis (ITSA) is regarded as the strongest quasi-experimental design in evaluating the impact of an intervention and is increasingly used in the evaluation of policy intervention in public health [27]. ITSA analyzes the difference in the trend and level before and after the intervention to explore the potential effect of policy intervention. The statistical model of a single interrupted time series analysis is as follows [28]:Y_t_ = β_0_ + β_1_T + β_2_X_t_ + β_3_TX_t_ + ε_t_(1)

T is a continuous variable denoting the number of months since the start of the study; X_t_ is a dummy variable that equals 1 before intervention and 0 after the intervention. β_0_ measures the level at the start of study, β_1_ measures the trend of the dependent variable before intervention, β_2_ measures the changes in the level of the dependent variable when the intervention occurred, and β_3_ represents the difference between the pre-intervention and post-intervention trends of the dependent variables. 

With the existence of a comparison group, the statistical model is as follows:Y_t_ = β_0_ + β_1_T + β_2_X_t_ + β_3_TX_t_ + β_4_Z + β_5_ZT + β_6_ZX_t_ + β_7_ZX_t_ T + ε_t_(2)

Z equals 1 for the intervention group and 0 for the comparison group. β_0_–β_3_ represents the estimation for the comparison group, which is similar to a single ITSA design, and β_4_–β_7_ measures the situation of the intervention group. β_4_ represents the difference between the comparison and intervention groups at the start of study; β_5_ represents the difference in the trend between the two groups prior to the intervention; β_6_ represents the difference between the two groups in the immediate change of the dependent variable after the intervention, and β_7_ represents the difference between the two groups in the change of the trend after the intervention [28,29,30]. 

Several requirements for ITSA analysis should be examined before regression: First, the time of the intervention should be clear [27]. In this article, September 2017 was chosen as the intervention time. Second, no agreements have been achieved on the requirement for the number of time points. However, some researchers argued that there should be at least 12 time points before and after the intervention while some proposed that six time points are needed before intervention and the total number of time points should exceed 12 [31]. In this article, we chose eight points before and nine points after the intervention to fulfil the requirements. Third, autocorrelation should be examined and adjusted using robust methods such as Newey–West SEs or Prais–Winsten SEs [32] with a time lag. We applied Stata 14.0 for the analysis and the Cumby–Huizinga test for the detection of autocorrelation, which was suggested by Linden et al. to examine the autocorrelation in time series [28]; the threshold was set to the 0.05 level [33]. The expenditures were converted to the 2017 dollar amount using the annual exchange rate. 

## 3. Results

### 3.1. Descriptive Analysis

Table 2 illustrates the procurement of negotiated medicine. The monthly average procurement expense of the three targeted anticancer drugs and Pegaspargase increased by 59.4%, 124.8%, 11.9%, and 54.1%, respectively. All negotiation anti-cancer drugs in the intervention groups revealed a significant increase in the procurement amount (calculated in DDDs). The monthly average procurement amount of rituximab, trastuzumab, and RHE increased by 160.5%, 520.1%, and 48.1%. The increase in the procurement amount exceeded the increase in expenditure, which revealed the primary effect of negotiation on the price discount. The monthly average procurement volume of Pegaspargase also increased by 55.2, which is basically in accordance with its increase in expenditure. The availability of trastuzumab increased by 7 percentage points while no changes were found for the availability of Rituximab or RHE. 

### 3.2. Impact of National Price Negotiation on Procurement Expenditure

Table 3 and Figure 1 illustrate the changes in the level and trend of the procurement expenditure, volume, and availability after the implementation of the policy. Compared with the comparison group, the monthly average procurement expenditure of Rituximab, Trastuzumab, and RHE increased by 6% (*p* < 0.1), 15% (*p* < 0.01), and 5% (*p* > 0.1), respectively, after the implementation of the negotiation in 11 provinces. No significant difference in level change of procurement expenditure was found between the intervention group and comparison group. The increasing trend of Trastuzumab before the intervention was significantly different from that of Pegaspargase (*p* < 0.05). 

For expenditure changes across eastern, middle, and western provinces, no significant differences were found between the expenditure of the intervention group and the comparison group of Rituximab and RHE. However, for Trastuzumab, the difference in the expenditure trend differed across regions (β_7_ = 0.07, *p* < 0.1 for eastern provinces; β_7_ = 0.28, *p* < 0.05 for middle provinces; β_7_ = 0.13, *p* < 0.1 for western region) (See Appendix A).

### 3.3. Impact of National Price Negotiation on Procurement Volume

Table 3 and Figure 2 show that compared with Pegaspargase, the monthly average procurement volume of Rituximab, Trastuzumab, and RHE increased by 99.51 DDDs (*p* < 0.1), 3673.17 DDDs (*p* < 0.01), and 30.73 DDDs (*p* > 0.1), respectively, after the implementation. The procurement volume of Rituximab increased immediately by 949.60 DDDs (*p* < 0.05) compared with the immediate change of the comparison group. No significant differences of level change were found for Trastuzumab and RHE respectively compared with control group. 

For procurement volume changes across difference regions, an immediate change only occurred in eastern regions for Rituximab (β_6_ = 2189.33, *p* < 0.01), and no differences in trends were observed. For Trastuzumab, an immediate change also occurred in eastern provinces only (β_6_ = 17,257.61, *p* < 0.01), similar to Rituximab. No significant differences in the level and trend changes were observed for RHE in all three regions (See Appendix A).

### 3.4. Impact of National Price Negotiation on Availability

Table 3 and Figure 3 reveal that the monthly average availability of Rituximab, Trastuzumab, and RHE increased by 0.24 (*p* < 0.1), 0.66 (*p* < 0.01), and 0.32 (*p* < 0.01) percentage points, respectively, compared with Pegaspargase. The immediate level change of availability for Rituximab and Trastuzumab increased by 1.56 percentage points (*p* < 0.1) and 5.24 percentage points respectively compared with Pegaspargase. 

For availability changes across different regions, significant differences were observed in eastern (β_6_ = 5.58, *p* < 0.01) and western (β_7_ = 0.71, *p* < 0.05) regions for Rituximab. For trastuzumab, level changes in eastern and middle provinces were 9.51 (*p* < 0.01) and 3.07 (*p* < 0.01) percentage points, respectively, while a trend change occurred only in the middle (β_7_ = 0.92, *p* < 0.01) and western (β_7_ = 0.89, *p* < 0.01) provinces compared with the comparison group. For RHE, there was a change in the trend in eastern (β_7_ = 0.45, *p* < 0.05) and western (β_7_ = 0.62, *p* < 0.05) provinces (See Appendix A). 

## 4. Discussion

Based on our analysis, the implementation of national drug price negotiation increased the volume of targeted anti-cancer drugs procured by each province, increased the availability and thus, increased the total expenditure of targeted anti-cancer drugs, especially for Rituximab and Trastuzumab. An increase in the procurement volume is usually seen in the pharmaceutical market after the inclusion of innovative drugs, especially anticancer drugs, in the reimbursement list [34], such as what happened in South Korea [35] and Mexico [36]. The impact of national drug price negotiations on the procurement volume is also consistent with current literature in China [19,21].

Our analysis also revealed a positive impact of the policy on the availability of negotiated anticancer drugs at a larger scale, which has not been fully investigated before in China [21]. A similar effect was reported when evaluating the effect of price negotiation on drug access in other countries, such as Mexico [36] and countries in western Europe [34,37]. A drug shortage is usually seen as one of the unintended consequences of policy lowering drug prices [38]. Although some reports in China suggested the existence of a drug shortage of cancer medicine because of national drug price negotiations and limits on total expenditure, our ITS analysis revealed the opposite. The changing behavior of doctors might be the main reason. After price reduction, patients and physicians who used to prescribe other medicines might begin to select the negotiated drugs, which were more expensive but not as expensive as before, and this has been observed in other studies [39,40]. 

However, we found that the policy might increase the total expenditure, while the current literature in China revealed the opposite or no significant changes [19,21]. The utilization of provincial procurement data rather than hospital panels might be the reason for the differences. The increasing expenditure might be the results of increased procurement volume by each province and increased availability within each province. In other words, the increase in volume has been the major influential factor in the expenditure change compared with a decrease in price, which is similar to cases in South Korea [35] and Mexico [36]. 

We also found that the impact of the policy differed across the eastern, middle, and western provinces and across drugs in China; the eastern provinces benefitted more compared with middle and western provinces. Several influential factors might contribute to the difference including the difference in physician’s treatment level, incidence of disease, average social–economic level of patients, and the baseline procurement situation of the anticancer drugs [21], which might have a negative impact on equity of access to targeted anti-cancer drugs in China. Further research at an individual level is warranted to investigate the utilization pattern of patients and doctors. 

Negotiation has been seen as an effective method to lower the drug price, improve equal access, and control drug expenditure [9], as well as to achieve balance between these goals, and therefore joint efforts of different stakeholder are warranted [41]. The increasing expenditure that originated from unmet medical needs should be promoted while those from inappropriate prescriptions should be prohibited [34]. For government, a range of strategies to control expenditure when including innovative drugs in the reimbursement list have been put forward and conducted, such as financial-based risk-sharing agreement (e.g., price–volume agreements in South Korea [35]) and performance-based risk-sharing agreements (e.g., coverage with evidence development in England), which have been widely applied in countries of Asia and Europe [42,43]. The Chinese government issued policies on the supervision of targeted anti-cancer drug utilization [44], but more improvement could be made by introducing more risk-sharing mechanisms in national drug price negotiation. The government and domestic industry can also cooperate in developing generics or biosimilars to further improve access and contain costs [34]. Scholars can bridge the gap between industry and government through assistance on value-based pricing and after-launch monitoring of cost-effectiveness by applying real world evidence. 

Different from current literature investigating the impact of national price negotiation in China, our analysis utilized nation-wide provincial level data and illustrated the impact of policy on volume, expenditure, and availability at a larger scale. Our analysis has the following limitations. First, out study is based on the analysis of aggregated provincial procurement data and we have no access to data at an individual level, which could investigate the potential existence of off-label usage of the three targeted anticancer drugs and further investigate the underlying reasons for different utilization patterns across different provinces and different drugs [20]. Second, we applied Pegaspargase as the intervention group because of the data limitation and the few anticancer drugs that were approved in China before 2017. The intervention group might not be perfect, and more drugs could be added as comparison groups if sufficient data exist. 

## 5. Conclusions

Based on a controlled ITS analysis, we found that the national drug price negotiation in China increased the procurement volume and expenditure of Rituximab and Trastuzumab and promoted the availability of all three anticancer drugs after controlling for potential co-intervention. The increasing expenditure at a provincial level might be the result of increasing the procured volume in each hospital and the increased availability of anticancer drugs. The impact of the policy differed across regions. Eastern provinces benefitted more compared with middle and western provinces, which could be the result of different social–economic factors. A joint effort from different stakeholders can further improve access to targeted anti-cancer drugs and contain costs by introducing risk-sharing mechanisms in negotiation and developing generics or biosimilars in China. Future effort could consider using individual-level data to capture the changing behavior of physicians and patients under the implementation of negotiation prices and the underlying reason for different utilization patterns after the implementation of negotiation prices for anticancer drugs. 

## Figures and Tables

**Figure 1 ijerph-19-04578-f001:**
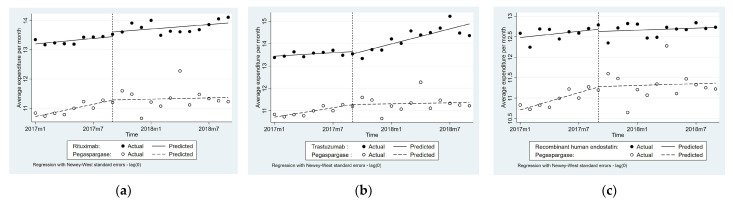
Average procurement expenditure (in 1000 RMB, log form) of three targeted anticancer drugs: (**a**) Rituximab; (**b**) Trastuzumab; (**c**) Recombinant human endostatin.

**Figure 2 ijerph-19-04578-f002:**
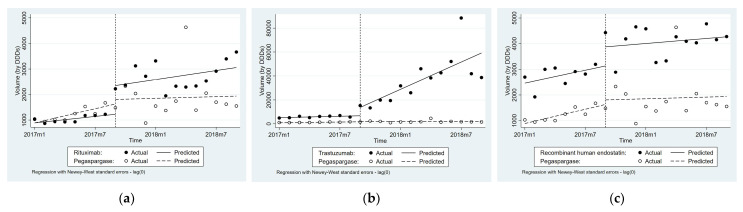
Average procurement volume based on DDDs of the three targeted anticancer drugs: (**a**) Rituximab; (**b**) Trastuzumab; (**c**) Recombinant human endostatin.

**Figure 3 ijerph-19-04578-f003:**
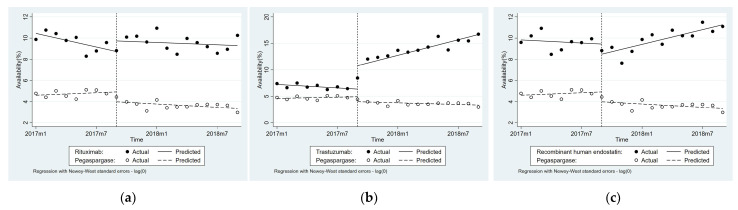
Availability of three targeted anticancer drugs: (**a**) Rituximab; (**b**) Trastuzumab; (**c**) Recombinant human endostatin.

**Table 1 ijerph-19-04578-t001:** Basic information of targeted anticancer drugs.

	Name	Indications	Launch Year in China
Intervention group	Rituximab	Non-Hodgkin’s lymphoma	2000
	Trastuzumab	Stomach cancer; Breast cancer	2002
	RHE	Lung cancer	2005
Comparison group	Pegaspargase	Acute Lymphocytic Leukemia	2009

**Table 2 ijerph-19-04578-t002:** Descriptive analysis for procurement of negotiated drugs.

Variables	Molecule	before September 2017	after September 2017	Diff (%)
Monthly average procurement expenditure (1000 US dollars)	Rituximab	573	912	59.4
	Trastuzumab	717	1611	124.8
	RHE	274	305	11.9
	Pegaspargase	55	84	54.1
Monthly average procurement volume	Rituximab	1036	2699	160.5
	Trastuzumab	5922	36,721	520.1
	RHE	2752	4075	48.1
	Pegaspargase	1210	1878	55.2
Availability (%)	Rituximab	10	10	0.0
	Trastuzumab	7	14	100.0
	RHE	9	10	11.1
	Pegaspargase	5	4	−20.0

**Table 3 ijerph-19-04578-t003:** Changes in the level and trend of the expenditure, volume, and availability for three targeted anticancer drugs and the comparison group.

Molecule	Variables	Expenditure (Log)	Volume (DDDs)	Availability (%)
Rituximab	Baseline difference	2.49 ***	2.92	5.73 ***
		(0.10)	(113.17)	(0.27)
	Baseline trend difference	−0.04	−51.39 *	−0.19 **
		(0.02)	(28.91)	(0.09)
	Difference in level change	0.15	949.60 **	1.56 *
		(0.23)	(415.79)	(0.92)
	Difference in trend change	0.06 *	99.51 *	0.24 *
		(0.03)	(57.66)	(0.13)
Tratuzumab	Baseline difference	2.73 ***	4343.00 ***	2.41 ***
		(0.09)	(336.95)	(0.28)
	Baseline trend difference	−0.04 *	101.57	−0.12 *
		(0.02)	(100.58)	(0.06)
	Difference in level change	−0.11	6765.52	5.24 ***
		(0.25)	(4375.29)	(0.85)
	Difference in trend change	0.15 ***	3673.17 ***	0.66 ***
		(0.04)	(1263.62)	(0.12)
Recombinant Human Endostatin	Baseline difference	1.77 ***	1574.03 ***	5.12 ***
		(0.14)	(295.96)	(0.57)
	Baseline trend difference	−0.05	−9.08	−0.08
		(0.03)	(57.21)	(0.09)
	Difference in level change	−0.06	563.87	0.59
		(0.23)	(538.47)	(0.70)
	Difference in trend change	0.05	30.73	0.32 ***
		(0.04)	(80.61)	(0.11)

Note: Standard errors in parentheses, (*** *p* < 0.01, ** *p* < 0.05, * *p* < 0.1).

## Data Availability

Data were collected and extracted from the China Drug Supply Information Platform. Corresponding authors have full access to the dataset used in the study and can help to achieve full access of the data of CDSIP under the permission of the National health commission in China.

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
