# Peer review of "Impacts of National Drug Price Negotiation on Expenditure, Volume, and Availability of Targeted Anti-Cancer Drugs in China: An Interrupted Time Series Analysis"

_ijerph, 2022, doi:10.3390/ijerph19084578_

Round 1

Reviewer 1 Report

The manuscripts can be improved, All three cancer drugs needs to be properly and also the authors needs to decide if they want include Pegaspargase as a fourth anti-cancer drug. The manuscript is not yet flowing nicely. 

Reviewer 2 Report

Title: Impacts of national drug price negotiation on expenditure, volume, and availability of targeted anti-cancer drugs in China: an interrupted time series analysis

MS ID: ijerph-1611537

The study examines drug price negotiation in China, which increased the expenditure and availability of some anti-cancer drugs. The study was well-conducted overall, with a clear objective and methodology.

Nonetheless, its presentation has many weaknesses that may lead to rejection if not addressed adequately.

I am going to give some examples and have some suggestions for further improvements if the authors are set to pursue this publication further.

First and foremost, the presentation needs to follow the standard reporting practices for a scientific paper. In many cases, the manuscript just uses abbreviations and acronyms without any explanations, and this practice could hardly be considered reasonable. For instance, nowhere in the manuscript “DDD” or “RHE” is explained.

It is the authors’ responsibility to check every detail like this to ensure the max readability and clarity to facilitate the audiences while reading a technical analysis.

Further example: The first paragraph of the discussion section is from the journal’s template; please remove it (line 205-208), which reads:

“Authors should discuss the results and how they can be interpreted from the perspective of previous studies and of the working hypotheses. The findings and their implications should be discussed in the broadest context possible. Future research directions may also be highlighted.”

Apart from the above, some issues will require your attention.

Public health issues are multifaceted. Changes in anti-cancer drugs’ prices involve many different actors, including the central government, local governments, hospitals, physicians, scientists, drug manufacturers, logistics, patients, monitoring agencies, and even the media covering such issues for the public. An integral framework for tackling a complex public health problem is necessary for consistently connecting related arguments. The authors should mention such a management framework in the discussion section. 

Additionally, studies on drug prices and related policies help provide an accurate assessment of respective expenditure, which is very important in terms of the national economy. However, misperceptions about costly scientific research can lead to insufficient governmental support and negatively affects long-term financial planning. The problem can be even more harmful in countries with relatively lower economic capacity. I think mentioning this issue alongside the direction for further studies would be a nice addition to the end of the discussion section. 

Reviewer 3 Report

I think this article has some value in that it gives us an insight into the current domestic drug pricing process in China. However, since previous studies [19,20] have shown that drug price negotiation interventions have an impact on patient cost reductions and volume increases, it is easy to see the same impact on each ministry. Since it is inferred, the novelty of the conclusion in this paper is lacking.

Some minor points are noted below.

In the introduction, an international comparison of drug prices converted by the U.S. dollar exchange rate/purchasing power parity is made, and China's position in this comparison frequently changes. Such wording is confusing to the reader and must be improved.

Regarding reference [7], the target information could not be found in the literature. Is it an error in the reference?

In Line 136, the Cumby-Huizinga test is used for the autocorrelation test, but it is necessary to state the reason for selecting this test method. In addition, three agents were selected as the comparison group and one agent was selected as the control, but the selection was not rational. Why not analyze the data in an integrated manner rather than individually in both groups?

There is a general lack of consideration regarding Conclusions. Many additions are needed based on your own results.

Round 2

Reviewer 1 Report

I recommend that the manuscript be accepted in its current form.

Author Response

Thank you for sincere suggestions! 

Reviewer 2 Report

Dear authors,

Thank you for your revised submission and the detailed response letter. Generally speaking, I find them both relevant and worthwhile.

I only have one issue left with the Conclusion, which still falls short of expectations. The issue the paper discusses has for long been a thorny one, therefore it will likely require both business and policy innovations. In light of this, I suggest the authors look into the discussion of the most recent and remarkable innovations during the Covid-19 times in the article: https://www.nature.com/articles/s41599-022-01034-6 and expand the Conclusion section to provide audiences with your thoughts. Without this, the paper sounds a bit "incomplete".

I wish you can handle this well and am, thus, delighted to render my recommendation "minor revision".

Best wishes,

Author Response

Thanks very much for taking your time to review this manuscript and we highly agree that the discussion and conclusion part still require improvements. We supplemented the cases in foreign countries, such as Norther Korea, Mexico and countries in western Europe and concluded that risk-sharing mechanism could be introduced in negotiation and generics or biosimilars could be developed under the joint effort of government and industry to further improve access of targeted anti-cancer drugs and containing their expenditure in both discussion and conclusion part.

Reviewer 3 Report

The authors were courteous in responding to my points and I am positive about the publication. However, I think there is a lack of consideration of the Conclusion section as before. Please provide a more generality, for example, including comparisons with other countries.

Author Response

Thank you for your sincere suggestions, please see the attachment
